# Topology Optimization for Digital Light Projector Additive Manufacturing Addressing the In-Situ Structural Strength Issue

**DOI:** 10.3390/polym15173573

**Published:** 2023-08-28

**Authors:** Jun Wang, Jikai Liu, Lei Li

**Affiliations:** 1Center for Advanced Jet Engineering Technologies (CaJET), Key Laboratory of High Efficiency and Clean Mechanical Manufacture, Ministry of Education, School of Mechanical Engineering, Shandong University, Jinan 250061, China; wang-jun@mail.sdu.edu.cn (J.W.); lei_li@sdu.edu.cn (L.L.); 2Key National Demonstration Center for Experimental Mechanical Engineering Education, Shandong University, Jinan 250061, China

**Keywords:** topology optimization, DLP 3D printing, design for additive manufacturing, stress constraint

## Abstract

A topology optimization approach is proposed for the design of self-supporting structures for digital light projector (DLP) 3D printing. This method accounts for the adhesion forces between the print part and the resin base during DLP printing to avoid failure of the part due to stress concentration and weak connections. Specifically, the effect of the process-related adhesion forces is first simulated by developing a design variable-interpolated finite element model to capture the intricate mechanical behavior during DLP 3D printing. Guided by the process model, a stress-constrained topology optimization algorithm is formulated with both the SIMP and RAMP interpolation schemes. The interpolations on the stress term and the design-dependent adhesion load are carefully investigated. A sensitivity result on the P-norm stress constraint is fully developed. Finally, the approach is applied to several 2D benchmark examples to validate its efficacy in controlling the process-caused peak P-norm stresses. The effects of alternating between the SIMP and RAMP interpolations and changing the stress upper limits are carefully explored during the numerical trials. Moreover, 3D printing tests are performed to validate the improvement in printability when involving the process-related P-norm stress constraint.

## 1. Introduction

Additive manufacturing reverses the conventional subtractive manufacturing process that slices the target part into layers and performs the manufacturing in a bottom-up layer-by-layer approach. The benefit of this novel approach is multifaceted: (i) complex-shaped parts can be manufactured similarly to their simple-shaped counterparts that decouple the proportional relationship between the manufacturing difficulty and the geometric complexity [1]; (ii) parts with tremendous interior cavities such as lattice structures that cannot be handled by conventional manufacturing are enabled for physical fabrication with additive manufacturing [2]; (iii) tedious tooling development (e.g., molds and fixtures) can be eliminated, which reduces the manufacturing preparation, saving both preparation time and costs; (iv) a variety of additive manufacturing processes have been developed and matured for industrial production that cover nearly all types of materials, including polymers, ceramics, metals and composites. The ISO/ASTM 52900:2015 has standardized the terminologies and principles of the diversified processes [3].

On the other hand, additive manufacturing is still a relatively expensive process compared to the conventional approaches due to the higher demands for raw materials and longer production cycles [4]. Hence, the success of applying additive manufacturing often leads to producing novel products dedicated to the layered manufacturing approach, i.e., equipped with extraordinary functionality or light weight by exploring the greatly expanded geometric design space that qualifies the costly additive production [4,5]. For the same reason, topology optimization is tightly connected to additive manufacturing, relying on algorithms and computational intelligence to achieve designs for additive manufacturing. The design results from topology optimization in general demonstrate excellent physical performance while utilizing only a limited quantity of materials, ensuring a light weight. Therefore, topology optimization for additive manufacturing has been of great significance in the past decade and interested readers can refer to a few dedicated review papers [6,7] for details.

Topology optimization for additive manufacturing takes into account the process characteristics to enhance the printability of the topological designs. The printability here can indicate a higher manufacturing success rate, lower manufacturing time or costs, a reduced gap between the as-designed and as-fabricated structural performance, etc. For instance, topology optimization for metal additive manufacturing takes the residual stress and distortion constraints into account [8,9,10,11], relieving the residual stress concentration to prevent cracks and reducing the *z*-axis warpage to avoid collisions to the powder brush (for powder bed fusion processes). Topology optimization for filament-based 3D printing processes, especially fused deposition modeling (FDM) for fiber-reinforced composites, includes both the topological variables and the path direction variables as the optimization targets [12,13,14], performing the concurrent optimization of the design with material anisotropies [15], which closes the gap between design and manufacture. Overhang-free topology optimization [16,17,18,19,20] controls the inclination angles of the overhanging features to eliminate the need for support structures, reducing the support producing time and simultaneously the cost [21,22]. The overhang-free optimization is generally applicable to all types of support-involved additive manufacturing processes. More issues, like the multi-scale design problem [23,24,25,26,27] and the post-machinability problem [28,29,30], have also been addressed.

Digital light projector (DLP), polymerizing the photosensitivity resins through the layered projection of UV light following the sliced 2D images, is a type of vat polymerization 3D printing process. The application scope of the DLP process is quite widespread since it is able to cure polymers, ceramics, and even metals with an excellent curing resolution [3]. The application cases for aerospace, aeronautics, and biomedical structures are widely reported. However, topology optimization for DLP 3D printing seems to be less discussed. Practically, the topology optimization results can be directly printed with the DLP machine and only sparse supports are required for process safety compared to other 3D printing techniques. On the other hand, process failure in DLP printing is frequently encountered with the repeatedly applied adhesion forces. Referring to Figure 1, the polymerized resin is bonded to the build bed but also glued to the tank base and, hence, adhesion forces take effect on the polymerized resin when separating it from the tank base to prepare the next layer for UV curing. Hence, the adhesion forces are applied to all layers and are likely to damage the part if weak connections exist. In summary, topology optimization for DLP 3D printing addressing the in-process structural strength issue deserves careful investigation.

To ensure the structural strength, stress-constrained topology optimization is generally performed. The anisotropic failure criteria [31,32] are applied to strength-based topology optimization due to the process-induced material anisotropy. Given the algorithm details under the density-based framework, the qp method [33,34] is widely adopted to prevent the stress singularity problem, i.e., eliminating high stress levels in the low-density elements. The P-norm or Kreisselmeier–Steinhauser function is applied to aggregate the many local stress constraints into one global peak stress constraint [35,36,37,38], and iterative adjustment to the P-norm approximation is also necessary to close the gap between the approximated and the exact peak stresses [39]. The other numerical issue is the high nonlinearity of the optimization problem, where the stress levels can be sharply perturbed by minor material distribution changes [35]. Hence, all parameters involved in configuring and solving the optimization problem should be fine-tuned. If performed under other topology optimization frameworks, the algorithm details would differ significantly. For instance, the black and white structural representation with the level set method avoids addressing the stress singularity issue [40,41,42]. The sharp solid-void transition for the Bidirectional Evolutionary Structural Optimization (BESO) method also eliminates this issue [43].

In this research, the stress-constrained topology optimization is configured differently from the majority of existing works. Specifically, apart from the work load, the process-induced load is applied in a layer-by-layer manner, indicating that, for every intermediate state, there is a coupled stress field to restrict the peak magnitude. Hence, multiple P-norm approximations on the peak stress are applied, each corresponding to a 3D printing intermediate state. Additionally, the process-induced load is design-dependent since more materials to be cured in the current layer will lead to a larger total force dragging the intermediate part. The stress-related problems involving design-dependent loads (pressure load or self-weight) are even more sophisticated [44,45,46], and Rational Approximation of Material Properties (RAMP) interpolation [47] is often used for this type of problem [8]. Hence, both the Solid Isotropic Material with Penalization (SIMP) and RAMP interpolations are investigated. More algorithm details are given in the following sections.

## 2. Topology Optimization Considerations for DLP 3D Printing

In the DLP 3D printing process, as shown in Figure 1, due to the upward motion of the print bed, adhesive forces are generated when separating the in-process workpiece from the tank base. Fractures or even complete failure may occur if weak structural connections exist, which should be avoided. Material design before printing can alleviate this issue [48], and, alternatively, topology optimization involving process-related stress constraints may completely eliminate this problem.

### 2.1. DLP Process Model

As illustrated in Figure 2, the domain Ω is divided into m layers, with each layer defined as ${Ω}_{i}$, where 1 ≤ i ≤ m. The bottom is fixed on the print bed, and the material curing process proceeds layer-by-layer in the bottom-up direction. Hence, the domain Ω is divided into three subdomains, ${Ω}_{d}$, ${Ω}_{e}$, and ${Ω}_{h}$, where ${Ω}_{d}$ is the collection of layers that have already been printed at the current stage, ${Ω}_{e}$ is the print layer that bears the adhesion forces in the current load step, and ${Ω}_{h}$ is the collection of layers that have not yet been printed.

During the printing process, the force-bearing layer ${Ω}_{e}$ is subjected to adhesion forces due to the lifting of the print platform and the detachment from the vat base. The adhesion forces act only on the polymerized materials and, hence, the force distribution is related to the design variables, serving as a design-dependent load and interpolated with the SIMP model specifically as
(1)$$\left\{\begin{array}{cc} \delta_{e}^{i}=1 & when\;{Ω}_{i}\in {Ω}_{e}\\ \delta_{e}^{i}=0 & otherwise \end{array}\right.$$
(2)$$f_{element}=\delta_{e}^{i}\rho_{e}^{3}ga$$
where $\;\delta_{e}^{i}$ is the indicator function representing domain ${Ω}_{e}$, $f_{element}$ is the total adhesion force applied to an element, g is the adhesion coefficient [49,50], and a is the length of the rectangular cell.

In topology optimization, equivalent nodal forces are required to represent the adhesion forces with the finite element method. Here, four-node quadrilateral elements are used for the finite element analysis and the equivalent nodal force at the *j*-th node of a four-node quadrilateral element can be obtained as
(3)$$f_{iy}^{e}=\int \delta_{e}^{i}N_{j}\rho_{e}^{3}gadA=\delta_{e}^{i}\rho_{e}^{3}ga\int N_{j}dA=\frac{1}{4}\delta_{e}^{i}\rho_{e}^{3}ga^{2}b$$
where $N_{j}$ is the shape function of the *j*-th node of the four-node rectangular element;$\;\rho_{e}$ is the physical density of the element; b is the width of the rectangular cell.

The progressive material curing process is simulated by configuring the element activation strategy in the finite element model. Specifically, ${Ω}_{h}$ is filled with substitute materials and will be activated sequentially in the bottom-up manner with the following indicator variable, $\;\theta_{e}^{i}$, which is the indicator function representing domain ${Ω}_{d}$.
(4)$$\left\{\begin{array}{cc} \theta_{e}^{i}=1 & when\;{Ω}_{i}\in {Ω}_{d}\\ \theta_{e}^{i}=0 & otherwise \end{array}\right.$$

When printing the *i*-th layer, the global stiffness matrix is the sum of the element stiffness matrices in ${Ω}_{d}$, which can be explicitly expressed using the indicator variable:(5)$$K_{i}=\overset{nele}{\underset{e=1}{\sum }}(\theta_{e}^{i}L_{e}^{T}K_{e}L_{e})$$

In this context, the matrix $L_{e}$ collects the nodal displacements $u_{e}\;$ of the *e*-th element from the global displacement vector U, satisfying the equation $u_{e}=L_{e}U$. $K_{e}\;$ represents the stiffness matrix of the element, and its calculation is conducted as follows:(6)$$K_{e}=\int B^{T}D_{e}Bd{Ω}_{e}=\eta_{e}^{E}\int B^{T}D_{0}Bd{Ω}_{e}$$

Here, $\eta_{e}^{E}$ represents the interpolation function for the elastic modulus, B denotes the strain–displacement matrix, and $D_{0}$ corresponds to the constitutive matrix of the solid material. The displacement $U_{i}$ due to the adhesion load at printing stage *i* is calculated as follows:(7)$$K_{i}U_{i}=F_{i}$$

In the implementation, multiple sliced layers are consolidated to eliminate the need for an extremely fine mesh and reduce the number of calls on finite element analysis.

### 2.2. AM Filter for Fabricable Self-Support Structures

A schematic diagram of the amplitude filter is provided in Figure 3 to illustrate its working principle. In this filter, the cells located in the lower layer with indices (*i* − 1, *j* − 1), (*i*, *j* − 1), and (*i* + 1, *j* − 1) are defined as the support region for cell (*i*, *j*). If there is no material in the support region, the amplitude filter will remove material from cell (*i*, *j*); otherwise, material is allowed to remain in cell (*i*, *j*). Through this rule, the amplitude filter achieves a 45° overhang self-supporting characteristic. The mathematical representation of the amplitude filter is shown below:(8)$$\xi_{i,j}=\left\{\begin{array}{c} \min\left({\bar{\rho }}_{i,j},\;\max\left({\bar{\rho }}_{i-1,j-1},{\bar{\rho }}_{i,j-1},{\bar{\rho }}_{i+1,j-1}\right)\right)\;\left(i,j\right)\in {Ω}_{u}\\ {\bar{\rho }}_{i,j}\;\left(i,j\right)\in {Ω}_{b} \end{array}\right.$$

The projection field $\bar{\rho }$ has its differential form expressed as follows:(9)$$\xi_{i,j}=smin\left({\bar{\rho }}_{i,j},\;smax\left({\bar{\rho }}_{i-1,j-1},{\bar{\rho }}_{i,j-1},{\bar{\rho }}_{i+1,j-1}\right)\right)\;\left(i,j\right)\in {Ω}_{u}$$

The operator “smin“ is defined as follows:(10)$$smin\left(a,b\right)=\frac{1}{2}\left(a+b-\sqrt{\left({\left(a-b\right)}^{2}+\epsilon_{s}\right)}+\epsilon_{s}{}^{2}\right)\;\left(i,j\right)\in {Ω}_{u}$$

The operator “smax” represents the maximum function “*P*-*Qmax*”, which is used to calculate the maximum value of elements within the support region.
(11)$$smax\left(a,b,c\right)=\sqrt[{Q}]{a^{P}+b^{P}+c^{P}}\;\left(i,j\right)\in {Ω}_{u}$$

The parameter $\epsilon_{s}=1\times 10^{-4}$ controls the approximation accuracy. The usage of P=40 and $Q=P+\frac{\log3}{\log\frac{1}{2}}$ is followed.

## 3. Interpolation Scheme and Total Stress Calculation

Density-based topology optimization conventionally uses interpolation schemes like Solid Isotropic Material with Penalization (SIMP) and Rational Approximation of Material Properties (RAMP) to relate the material properties to the element densities (topological variable). This study employs both the SIMP and RAMP interpolations to gain insights into their differences and similarities for adhesion-aware topology optimization. SIMP uses a power-law function to penalize intermediate densities, while RAMP uses a rational function to provide a smooth transition between solid and void phases. We implement both schemes in this work to compare their performance in mitigating adhesion-induced stress concentration, a key challenge in DLP-type additive manufacturing. The results will disclose the benefits and limitations of both interpolations in handling in-situ structural strength constraints in topology optimization for polymer structures made by DLP 3D printing. The specific interpolation schemes are given below:(12)$$E_{e}=E_{min}+\eta_{e}^{E}\left(E_{0}-E_{min}\right)$$

SIMP: $\eta_{e}^{E}=\rho_{e}{}^{p\_SIMP}$

RAMP: $\eta_{e}^{E}=\frac{\rho_{e}}{1+\left(1-\rho_{e}\right)p\_RAMP}$

$E_{e}$ is the interpolated elastic modulus for element *e*; p_SIMP is the penalty parameter of the *SIMP* interpolation; p_RAMP is the *RAMP* interpolation penalty parameter; $\rho_{e}$ is the physical density of element *e*; $E_{0}$ represents the elastic modulus of the solid material; $E_{min}$ is the elastic modulus for voids.

The stress vector $\sigma_{e}$ for element *e* is calculated through
(13)$$\sigma_{e}=\theta_{e}^{i}D_{0}B_{e}u_{e}$$
where $D_{0}$ is the stiffness matrix of the solid material; $B_{e}$ is the strain–displacement matrix; $u_{e}$ is the local displacement vector. The two-dimensional stress vector $\sigma_{e}$ is
(14)$$\sigma_{e}={\left(\sigma_{ex},\sigma_{ey},\tau_{exy}\right)}^{T}$$

The relaxed stress measure ${\hat{\sigma }}_{e}\left(\rho_{e}\right)$ is expressed as
(15)$${\hat{\sigma }}_{e}\left(\rho_{e}\right)=\eta \left(\rho_{e}\right)\sigma_{e}$$

A general stress punishment scheme is given as
(16)$$\eta \left(\rho_{e}\right)={\left(\rho_{e}\right)}^{q}$$
where q is a non-negative stress relaxation parameter. For full material cells, ${\hat{\sigma }}_{e}$ is equal to $\sigma_{e}$, and for an empty cell, it satisfies
(17)$$\underset{\rho_{e}\to 0}{\lim}{\hat{\sigma }}_{e}\left(\rho_{e}\right)=0$$

The two-dimensional von Mises stress is defined by
(18)$${\hat{\sigma }}_{vm,e}={\left(\sigma_{ex}{}^{2}+\sigma_{ey}{}^{2}-\sigma_{ex}\sigma_{ey}+3\tau_{exy}^{2}\right)}^{\frac{1}{2}}$$

A standard P-norm global stress measurement is used to approximate the maximum stress as follows:(19)σPN=∑e=1neleσ^vm,ep1p
where ${\hat{\sigma }}_{vm,e}$ is the von Mises stress at the centroid of element *e*, and p is the P-norm aggregation parameter. As p→∞, the P-norm approaches the maximum value of ${\hat{\sigma }}_{vm}$.
(20)Maxσ^vm≤∑e=1neleσ^vm,ep1p

In general, a larger p provides a more accurate approximation of the maximum von Mises stress. However, if p is excessively large, it can lead to an ill-posed problem and cause severe oscillations during the optimization process. Therefore, it is important to select an appropriate p value to ensure smooth convergence while sufficiently approximating the maximum stress.

## 4. Formulation and Solution of Optimization Problems

### 4.1. Optimization Problem Formulation

The objective function of the optimization problem is to minimize the compliance of the structure under the maximum stress constraint and volume constraint, and the mathematical expression of the problem is as follows:$$\left\{\begin{array}{c} find:\;\rho \\ minimize:C=U_{c}^{T}K_{c}U_{c}\\ subject\;to:\left\{\begin{array}{c} K_{i}U_{i}=F_{i}\left(i=1,2,\ldots ,m\right)\\ K_{c}U_{c}=F_{c}\\ \frac{V\left(\rho \right)}{V_{0}}=f\\ \sigma_{PN}^{i}\leq \sigma \\ 0<\rho_{min}\leq \forall \rho_{e}\leq 1 \end{array}\right. \end{array}\right.$$

In this equation, $U_{c}$ represents the nodal displacement field induced by mechanical external loads, $K_{c}$ denotes the overall stiffness matrix of the structure, and $V\left(\rho \right)=\sum_{e=1}^{nele}{\overset{=}{\rho }}_{e}$ represents the material volume. $V_{0}$ denotes the total volume of the design domain. Setting $\rho_{min}$ to 1 × 10^−3^ helps to avoid matrix singularity during layer-by-layer strength analysis and static mechanical analysis. 

### 4.2. Sensitivity Analysis

To conduct stress-based topology optimization, it is necessary to provide the sensitivities of the global stress measure on the element densities. The Method of Moving Asymptotes (MMA) [51] is employed to solve the optimization problem, which requires the first-order sensitivity information of the objective function and constraints. The gradient $\sigma_{PN}^{i}$ is calculated using the chain rule as follows:(21)$$\frac{\partial \sigma_{PN}^{i}}{\partial \rho_{j}}=\frac{\partial \sigma_{PN}^{i}}{\partial \overset{=}{\rho }}\;\frac{\partial \overset{=}{\rho }}{\partial \bar{\rho }}\frac{\partial \bar{\rho }}{\partial \tilde{\rho }}\frac{\partial \tilde{\rho }}{\partial \rho_{j}}$$

In the equation, $\overset{=}{\rho }$ represents the physical density field after the three layers of filtering projection. $\frac{\partial \overset{=}{\rho }}{\partial \bar{\rho }}$, $\frac{\partial \bar{\rho }}{\partial \tilde{\rho }}$, and $\frac{\partial \tilde{\rho }}{\partial \rho_{j}}$, respectively, denote the standard modifications of sensitivity due to the AM filter, Heaviside projection filter, and smoothing filter.

For $\frac{\partial \sigma_{PN}^{i}}{\partial \overset{=}{\rho }}$, we can obtain
(22)$$\frac{\partial \sigma_{PN}^{i}}{\partial \overset{=}{\rho }}=\sum_{e=1}^{nele}\left(\frac{\partial \sigma_{PN}^{i}}{\partial {\hat{\sigma }}_{vm,e}}\left[{\left(\frac{\partial {\hat{\sigma }}_{vm,e}}{\partial {\hat{\sigma }}_{e}}\right)}^{T}\frac{\partial \eta \left({\overset{=}{\rho }}_{e}\right)\sigma_{e}}{\partial \overset{=}{\rho }}\right]\right)\;=\sum_{e=1}^{nele}\left(\frac{\partial \sigma_{PN}^{i}}{\partial {\hat{\sigma }}_{vm,e}}\left[{\left(\frac{\partial {\hat{\sigma }}_{vm,e}}{\partial {\hat{\sigma }}_{e}}\right)}^{T}\frac{\partial \eta \left({\overset{=}{\rho }}_{e}\right)}{\partial \overset{=}{\rho }}\sigma_{e}\right]\right)+\sum_{e=1}^{nele}\left(\frac{\partial \sigma_{PN}^{i}}{\partial {\hat{\sigma }}_{vm,e}}\left[{\left(\frac{\partial {\hat{\sigma }}_{vm,e}}{\partial {\hat{\sigma }}_{e}}\right)}^{T}\eta \left({\overset{=}{\rho }}_{e}\right)\frac{\partial \sigma_{e}}{\partial \overset{=}{\rho }}\right]\right)$$

The above equation can be written in the following form:(23)$$\left\{\begin{array}{c} \frac{\partial \sigma_{PN}^{i}}{\partial \rho_{j}}=T_{1}+T_{2}\\ T_{1}=\sum_{e=1}^{nele}\left(\frac{\partial \sigma_{PN}^{i}}{\partial {\hat{\sigma }}_{vm,e}}\left[{\left(\frac{\partial {\hat{\sigma }}_{vm,e}}{\partial {\hat{\sigma }}_{e}}\right)}^{T}\frac{\partial \eta \left({\overset{=}{\rho }}_{e}\right)}{\partial \overset{=}{\rho }}\sigma_{e}\right]\right)\\ T_{2}=\sum_{e=1}^{nele}\left(\frac{\partial \sigma_{PN}^{i}}{\partial {\hat{\sigma }}_{vm,e}}\left[{\left(\frac{\partial {\hat{\sigma }}_{vm,e}}{\partial {\hat{\sigma }}_{e}}\right)}^{T}\eta \left({\overset{=}{\rho }}_{e}\right)\frac{\partial \sigma_{e}}{\partial \overset{=}{\rho }}\right]\right) \end{array}\right.$$

For $\frac{\partial \sigma_{PN}^{i}}{\partial {\hat{\sigma }}_{vm,e}}$, it can be expressed as follows:(24)$$\frac{\partial \sigma_{PN}^{i}}{\partial {\hat{\sigma }}_{vm,e}}=\sum_{e=1}^{nele}\left({\left({\left(\partial {\hat{\sigma }}_{vm,e}\right)}^{p}\right)}^{\frac{1}{p}-1}{\left(\partial {\hat{\sigma }}_{vm,e}\right)}^{p-1}\right)$$

The derivative of the element von Mises stress with respect to its stress components can be expressed as
(25)$$\left\{\begin{array}{c} \frac{\partial {\hat{\sigma }}_{vm,e}}{\partial {\hat{\sigma }}_{ex}}=\frac{1}{2{\hat{\sigma }}_{vm,e}}\left(2{\hat{\sigma }}_{ex}-{\hat{\sigma }}_{ey}\right)\\ \frac{\partial {\hat{\sigma }}_{vm,e}}{\partial {\hat{\sigma }}_{ey}}=\frac{1}{2{\hat{\sigma }}_{vm,e}}\left(2{\hat{\sigma }}_{ey}-{\hat{\sigma }}_{ex}\right)\\ \frac{\partial {\hat{\sigma }}_{vm,e}}{\partial {\hat{\tau }}_{exy}}=\frac{3}{{\hat{\sigma }}_{vm,e}}{\hat{\tau }}_{exy} \end{array}\right.$$

The $\frac{\partial \eta \left({\overset{=}{\rho }}_{e}\right)}{\partial \overset{=}{\rho }}$ is simply obtained as
(26)$$\frac{\partial \eta \left({\overset{=}{\rho }}_{e}\right)}{\partial \overset{=}{\rho }}=q{\overset{=}{\rho }}_{e}{}^{q-1}\frac{\partial {\overset{=}{\rho }}_{e}}{\partial \overset{=}{\rho }}$$

The analytic form of $\frac{\partial \sigma_{e}}{\partial \overset{=}{\rho }}$ can be expressed as
(27)$$\frac{\partial \sigma_{e}}{\partial \overset{=}{\rho }}=\theta_{e}^{i}D_{0}B_{e}\frac{\partial u_{e}}{\partial \overset{=}{\rho }}=\theta_{e}^{i}D_{0}B_{e}L_{e}\frac{\partial U_{i}}{\partial \overset{=}{\rho }}$$

Then, we obtain the expression of $T_{2}$ through substitutions
$$T_{2}=\sum_{e=1}^{nele}\left(\frac{\partial \sigma_{PN}^{i}}{\partial {\hat{\sigma }}_{vm,e}}\left[{\left(\frac{\partial {\hat{\sigma }}_{vm,e}}{\partial {\hat{\sigma }}_{e}}\right)}^{T}\eta \left({\overset{=}{\rho }}_{e}\right)\frac{\partial \sigma_{e}}{\partial \overset{=}{\rho }}\right]\right)$$
(28)$$=\sum_{e=1}^{nele}\left(\frac{\partial \sigma_{PN}^{i}}{\partial {\hat{\sigma }}_{vm,e}}{\left(\frac{\partial {\hat{\sigma }}_{vm,e}}{\partial {\hat{\sigma }}_{e}}\right)}^{T}\eta \left({\overset{=}{\rho }}_{e}\right)\theta_{e}^{i}D_{0}B_{e}L_{e}\frac{\partial U_{i}}{\partial \overset{=}{\rho }}\right)$$

Given the unknown $\frac{\partial U_{i}}{\partial \overset{=}{\rho }}$, we take the derivatives on the governing equation and derive
(29)$$\frac{\partial K_{i}}{\partial \overset{=}{\rho }}U_{i}+K_{i}\frac{\partial U_{i}}{\partial \overset{=}{\rho }}=\frac{\partial F_{i}}{\partial \overset{=}{\rho }}$$

We solve the above formula and then we have
(30)$$\frac{\partial U_{i}}{\partial \overset{=}{\rho }}=K_{i}{}^{-1}\left(\frac{\partial F_{i}}{\partial \overset{=}{\rho }}-\frac{\partial K_{i}}{\partial \overset{=}{\rho }}U_{i}\right)$$

Thus, $T_{2}$ can be expressed as
(31)$$T_{2}=\sum_{e=1}^{nele}\left(\frac{\partial \sigma_{PN}^{i}}{\partial {\hat{\sigma }}_{vm,e}}{\left(\frac{\partial {\hat{\sigma }}_{vm,e}}{\partial {\hat{\sigma }}_{e}}\right)}^{T}\eta \left({\overset{=}{\rho }}_{e}\right)\theta_{e}^{i}D_{0}B_{e}L_{e}K_{i}{}^{-1}\left(\frac{\partial F_{i}}{\partial \overset{=}{\rho }}-\frac{\partial K_{i}}{\partial \overset{=}{\rho }}U_{i}\right)\right)$$
where $\frac{\partial F_{i}}{\partial \overset{=}{\rho }}$ is obtained from the previous calculation,
(32)$$\frac{\partial F_{i}}{\partial \overset{=}{\rho }}=\overset{nele}{\underset{e=1}{\sum }}\frac{3}{4}\delta_{e}^{i}{\overset{=}{\rho }}_{e}^{2}\frac{\partial {\overset{=}{\rho }}_{e}}{\partial \overset{=}{\rho }}ga^{2}bL_{e}^{T}$$

For $\frac{\partial K_{i}}{\partial \overset{=}{\rho }}$, it can be expressed as
(33)$$\frac{\partial K_{i}}{\partial \overset{=}{\rho }}=\sum_{e=1}^{nele}\theta_{e}^{i}L_{e}^{T}\frac{\partial K_{e}}{\partial \overset{=}{\rho }}L_{e}=\sum_{e=1}^{nele}\theta_{e}^{i}L_{e}^{T}\left(\frac{\partial \eta_{e}^{E}}{\partial {\overset{=}{\rho }}_{e}}\frac{\partial {\overset{=}{\rho }}_{e}}{\partial \overset{=}{\rho }}K_{0}\right)L_{e}$$

In the formula, $\eta_{e}^{E}$ is the elastic modulus interpolation function, and $K_{0}$ is the element stiffness matrix in full solid state. The detailed expression of the element stiffness is given by the following formula:(34)$$K_{0}=\int B^{T}D_{0}Bd{Ω}_{e}$$

## 5. Numerical Examples

In this section, we employ two classic two-dimensional cases, the MBB beam and cantilever beam, to validate the efficacy of the proposed method. In all numerical implementations, the “45-degree rule” is adopted, i.e., the maximum self-supporting overhang angle is set as 45 degrees. Finite element analysis is performed using four-node quadrilateral elements. The MMA optimizer is utilized as the solver with a move limit of 0.1. The sharpness parameter for the Heaviside projection filter starts from 1, doubles for every 50 iterations, and ends with a maximum value of 32.

Regarding the implementation specifications, finite element analysis is performed on the general minimum compliance topology optimization result with only the volume constraint and no stress constraint. The stress fields due to adhesion loads are simulated to disclose the prone-to-failure features and provide references for setting the stress restriction upper bounds. Thereafter, the stress-constrained and volume-constrained minimum compliance topology optimization is carried out to design the process-safe optimal structures. As stated earlier, setting q to 0.5 can help to avoid gray elements remaining in the optimized structure for the traditional stress-constrained SIMP setup. However, artificial high stresses are likely to appear at the boundary non-solid elements since the AM filter is applied after Heaviside projection and the approximated smax and smin operators unavoidably lead to non-solid elements. q=0.5 cannot eliminate these non-solid effects; instead, it artificially magnifies the local stresses that disturb the convergence. Hence, in the following case study, q is set to 2 to prevent non-physical stress concentration.

### 5.1. The MBB Beam Structure

The first case is an MBB beam example, as illustrated in Figure 4. Due to the symmetry condition, only the right half of the structure is considered in optimization with the discretization of nelx=180 and nely=60. A concentrated vertical force (F=1) is loaded at the top left corner of the half domain, while the bottom right corner is supported on a roller. The left edge adopts the symmetry boundary condition. The element size of 1 × 1 is adopted. The volume fraction upper limit is chosen to be 0.5. The filter radius rmin is set to 5. Other parameters are selected as follows: the *SIMP* interpolation parameter p_SIMP is 5; the *RAMP* interpolation parameter p_RAMP is 20; the P-norm parameter p is 10.

Using SIMP interpolation, the minimum compliance of 194.8950 is first obtained without applying the stress constraint, and if we apply the layer-based adhesion load to this result, the maximum P-norm stress occurs at the 49th layer’s adhesion load, reaching 14.9651. Hence, to reduce the risk of failure, we take the stress limits of 10, 5, and 3, respectively, to perform the process of stress-constrained topology optimization, obtaining the optimized structures and corresponding stress profiles for the most stress-concentrated load steps, as presented in Figure 5. Note that the full MBB structure is demonstrated instead of only a symmetric half.

With the P-norm stress limit of 10, the optimized structural compliance is 194.4829 and the maximum P-norm stress occurs at the 49th layer’s load step, reduced to 8.4181.

With the P-norm stress limit of 5, the optimized structural compliance is 208.4969 and the maximum P-norm stress occurs at the 37th layer’s load step, reduced to 4.2599.

With the P-norm stress limit of 3, the optimized structural compliance is 222.9046 and the maximum P-norm stress occurs at the 53rd layer’s adhesion load, reduced to 3.0000.

The convergence curves under the three P-norm stress constraint conditions are shown in Figure 6.

From the iteration curves in Figure 6, we can see that the convergence of the iteration curves is difficult with the SIMP interpolation. The curves fluctuate violently and the fluctuation intensity increases with the enhanced strength requirement.

Using RAMP interpolation, the minimum compliance of 187.7462 is obtained without the stress constraint, and if we apply the layer-based adhesion load, the maximum P-norm stress occurs at the 47th layer’s adhesion load, reaching 18.3128. Hence, to reduce the failure risk, we again take the P-norm stress limits of 10, 5, and 3, respectively, to perform the process of stress-constrained topology optimization, obtaining the optimized structures and corresponding stress profile plots for the most stress-concentrated load steps, as shown in Figure 7.

With the stress limit of 10, the optimized structural compliance is 186.1907 and the maximum P-norm stress occurs at the 47th layer’s adhesion load with the peak value of 8.7847.

With the stress limit of 5, the optimized structural compliance is 187.4594 and the maximum P-norm stress occurs at the 50th layer’s adhesion load with the peak value of 4.9952.

With the stress limit of 3, the optimized structural compliance is 190.3872 and the maximum P-norm stress occurs at the 50th layer’s adhesion load with the peak value of 2.9935.

Figure 8 summarizes the optimized structural compliances subject to different constraint conditions and interpolation methods. It can be seen that under the currently selected parameters, using RAMP interpolation leads to structures with better overall stiffness compared to SIMP interpolation. However, the difference between the two interpolation methods may be decreased by adjusting the interpolation details.

The convergence curves under the three stress constraint conditions for the RAMP interpolation scenario are shown in Figure 9.

From Figure 9, we can see that the iteration curves converge faster with the RAMP interpolation, and fewer fluctuations are observed than in Figure 6. Additionally, the overall stiffness deterioration due to involving stress constraints is moderate for the RAMP interpolation.

### 5.2. Cantilever Beam

The second case is performed on a cantilever beam, as illustrated in Figure 10, with nelx=160 and nely=80. The load is applied on the bottom right corner, while the left edge is completely fixed. The volume fraction upper limit is chosen to be 0.5. The filter radius $r_{min}$ is set to 5. Other parameters are selected as follows: the SIMP interpolation parameter p_SIMP is 5; the RAMP interpolation parameter p_RAMP is 20; the P-norm parameter p is 10.

(1)Using SIMP interpolation, the minimum compliance of 81.5803 is first obtained without imposing the stress constraint, and if we apply the layer-based adhesion load to this result, the maximum P-norm stress occurs at the 59th layer’s adhesion load, reaching 17.906. Hence, to reduce the risk of failure, we take the stress limits of 10, 5, and 3, respectively, to perform the process of stress-constrained topology optimization, obtaining the optimized structures and corresponding stress profiles for the most stress-concentrated load steps, referring to Figure 11.

With the P-norm stress limit of 10, the optimized structural compliance is 76.3174 and the maximum P-norm stress of 10.0000 occurs at the 45th layer’s adhesion load step.

With the P-norm stress limit of 5, the optimized structural compliance is 76.5998 and the maximum P-norm stress of 4.9990 occurs at the 60th layer’s adhesion load step.

With the P-norm stress limit of 3, the optimized structural compliance is 79.3136 and the maximum P-norm stress of 3.0000 occurs at the 47th layer’s adhesion load step.

The convergence curves under the three stress constraint conditions are given in Figure 12.

From the iteration curves in Figure 12, we can see that when using SIMP interpolation, compared to the MBB beam case, the iteration curves are smoother and converge more easily. The stiffness of the optimized results gradually deteriorates with the increased strength requirement, but the overall stiffness change is small for the stress limits of 10 and 5. It is unexpected that the optimization result without applying the stress constraint has the lowest stiffness, possibly due to the local optimum issue.

(2)Using RAMP interpolation, the minimum compliance of 68.2931 is obtained without the stress constraint, and if we apply the layer-based adhesion load, the maximum stress occurs at the 63rd layer’s adhesion load step, reaching 17.1274. Hence, to reduce the failure risk, we again take the stress limits of 10, 5, and 3, respectively, to perform the process of stress-constrained topology optimization, obtaining the optimized structures and corresponding stress profiles for the most stress-concentrated load steps, shown in Figure 13.

With the P-norm stress limit of 10, the optimized structural compliance is 68.6842 and the maximum P-norm stress of 9.8595 occurs at the 62nd layer’s adhesion load.

With the P-norm stress limit of 5, the optimized structural compliance is 68.7421 and the maximum P-norm stress of 4.9990 occurs at the 64th layer’s adhesion load.

With the P-norm stress limit of 3, the optimized structural compliance is 69.7818 and the maximum P-norm stress of 2.9969 occurs at the 65th layer’s adhesion load.

Figure 14 summarizes the optimized structural compliances subject to different constraint conditions and interpolation methods. The observation remains the same as the MBB beam case, namely that under the current parameter selection, the RAMP interpolation produces structures with better global stiffness.

The convergence curves under the three stress constraint conditions are given in Figure 15. Similar to the first numerical example, the stiffness condition of the structure gradually deteriorates with the increased strength requirement, and the overall stiffness deterioration is evidently milder than with the SIMP interpolation.

## 6. Experimental Validation

In order to validate the efficacy of the proposed method, an experimental test is designed and carried out in this section. An MBB beam with an 8:1 aspect ratio is optimized and manufactured for this experimental investigation, since the increased structure details provide a higher possibility of failure. The stress-constrained topology optimization is performed with SIMP interpolation and the following parameters: nelx=240, nely=60, 50% volume fraction upper limit, SIMP penalization factor p_SIMP=4, filter radius of 4, P-norm parameter p=10, stress interpolation factor q=2, 60 layers of adhesion load, and the MMA optimizer move limit of 0.1.

Finite element analysis on the topology optimization result without the stress constraint shows the maximum P-norm stress occurring at the 52nd layer’s adhesion load, reaching 38.01. The overall compliance of the structure is 399.5586. The 3D model for additive manufacturing obtained after post-processing the optimization result is shown in Figure 16.

Then, we set a process stress upper limit of 2 and obtain the optimized design as shown in Figure 17. The maximum stress occurs at the 34th layer’s adhesion load, having the peak P-norm stress of 2.000. The overall structural compliance is 412.1184, slightly increased compared to the non-constrained design in Figure 15. However, the weak and stress-concentrated thin struts are successfully removed in this new design.

As shown in Figure 18, a thickness of 0.5 mm is selected for model generation. Three repeated of print attempts are made for each of the above two compared designs. The print results are demonstrated in Figure 19. It is observed that the regular topology optimization designs all fail at the stress-concentrated thin strut, while the process stress-constrained designs are all successfully printed with complete structural details. Therefore, the proposed method enhances the in-process structural strength and can resolve the issue of print failure due to the repeating adhesion loads.

## 7. Conclusions

This paper has developed a topology optimization design method for DLP additive manufacturing that addresses the process-related structural strength issue, leading to the following conclusions.

(1)To mimic the in-process stress concentrations, an SIMP-like interpolation is proposed to simulate the adhesion forces between the workpiece and resin base for DLP additive manufacturing.(2)A greater than 1 penalization parameter is adopted for the stress term to prevent artificial high stresses at the boundary non-solid elements since the AM filter is applied after Heaviside projection and the approximated
smax and
smin operators unavoidably lead to non-solid elements.(3)Both the SIMP and RAMP interpolations allow us to achieve topology optimization designs considering the maximum volume fraction and P-norm stress constraints. SIMP interpolation exhibits more fluctuations in converging the optimization process. RAMP interpolation has a smoother convergence process, although some cases show a small amount of irremovable gray elements.(4)Experiments validate the necessity and effect of imposing the stress constraints. The stress-concentrated thin struts are replaced by thickened structural features through size and shape enhancement, and the overall stress field tends to distribute more evenly. Hence, failures are not encountered in the modified designs and, simultaneously, the local shaping accuracy is guaranteed as well.(5)The layerwise P-norm stresses are summarized in Figure 20 for both the SIMP and RAMP interpolations. Apparently, a stricter stress restriction leads to an overall reduction in P-norm stresses, indicating the effectiveness of the layerwise stress constraints.(6)A stricter stress restriction, i.e., lower P-norm stress upper limit, causes the decreased overall stiffness of the optimized structure for both the SIMP and RAMP interpolations.(7)Interestingly, topological optimization that considers the maximum volume fraction and stress constraints may yield better structural stiffness than topological optimization that only considers the volume constraints. This phenomenon deserves careful further investigation.(8)The build orientation plays an equally important role in topology optimization design considering the process stress constraints in additive manufacturing. Appropriate build directions may allow us to resolve the maximum P-norm stress constraints without significantly impacting the structural stiffness. This topic will be investigated in our forthcoming research.(9)Finally, considering the computational cost, the finite element analysis simulating the layer-based adhesion loading conditions and the stress term-related adjoint equations occupies the majority of the calculation time. Adopting parallel computing could greatly reduce the computing time, especially for larger-scale problems, and this scheme deserves further exploration as well.

## Figures and Tables

**Figure 1 polymers-15-03573-f001:**
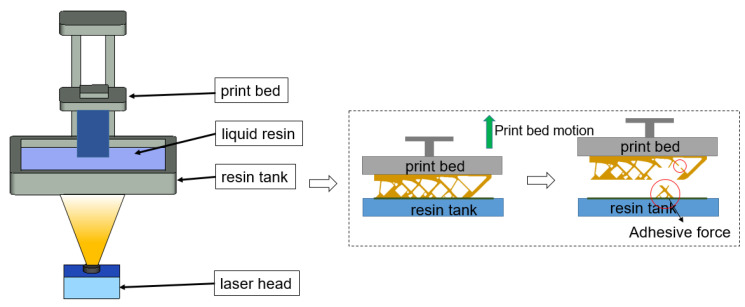
Self-supporting structure failure in DLP additive manufacturing process.

**Figure 2 polymers-15-03573-f002:**
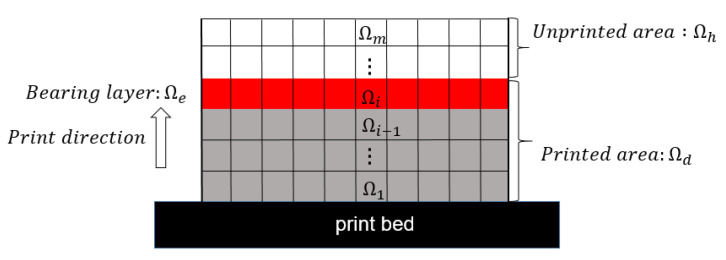
Additive manufacturing process model.

**Figure 3 polymers-15-03573-f003:**
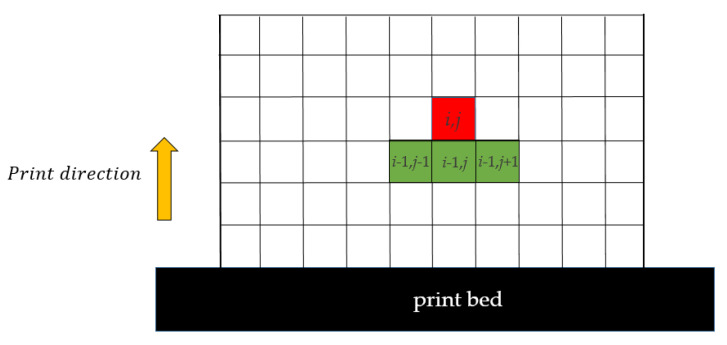
The schematic diagram of the AM filter for 2D cases.

**Figure 4 polymers-15-03573-f004:**
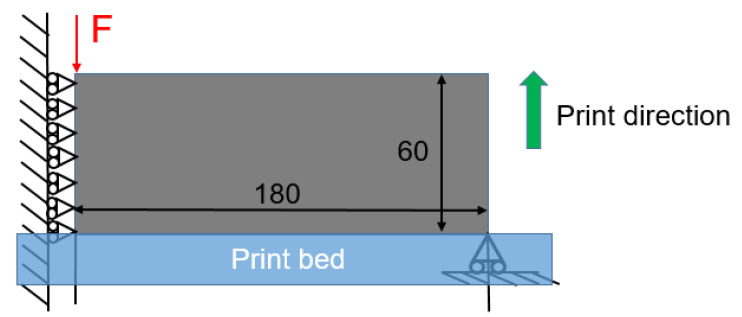
Boundary condition for the MBB beam.

**Figure 5 polymers-15-03573-f005:**
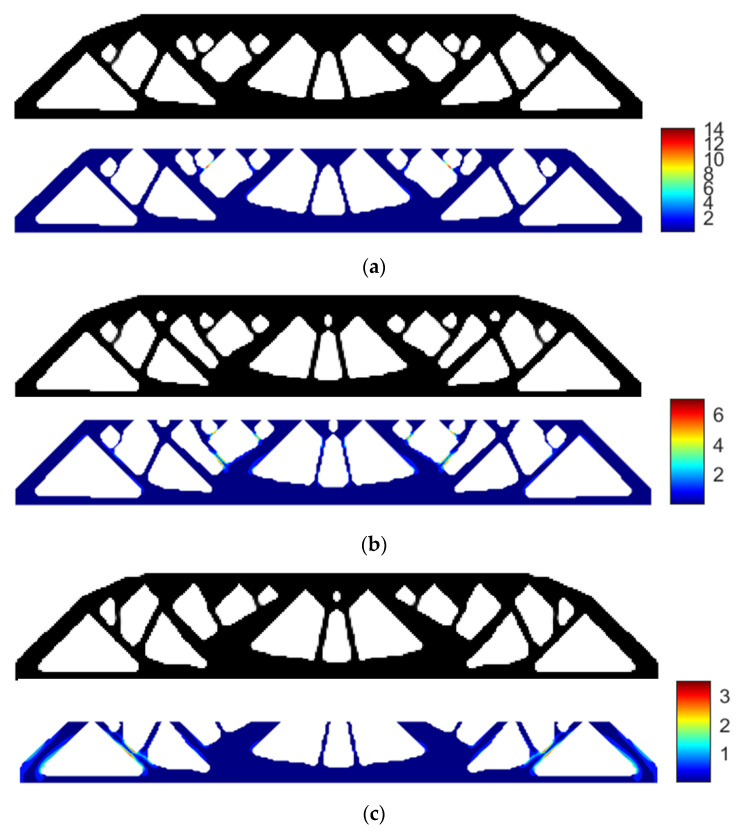
The optimized MBB beam structures and the stress profile plots corresponding to the most stress-concentrated load steps obtained with SIMP interpolation: (**a**) without P-norm stress constraint; (**b**) P-norm stress limit set to 10; (**c**) P-norm stress limit set to 5; (**d**) P-norm stress limit set to 3.

**Figure 6 polymers-15-03573-f006:**
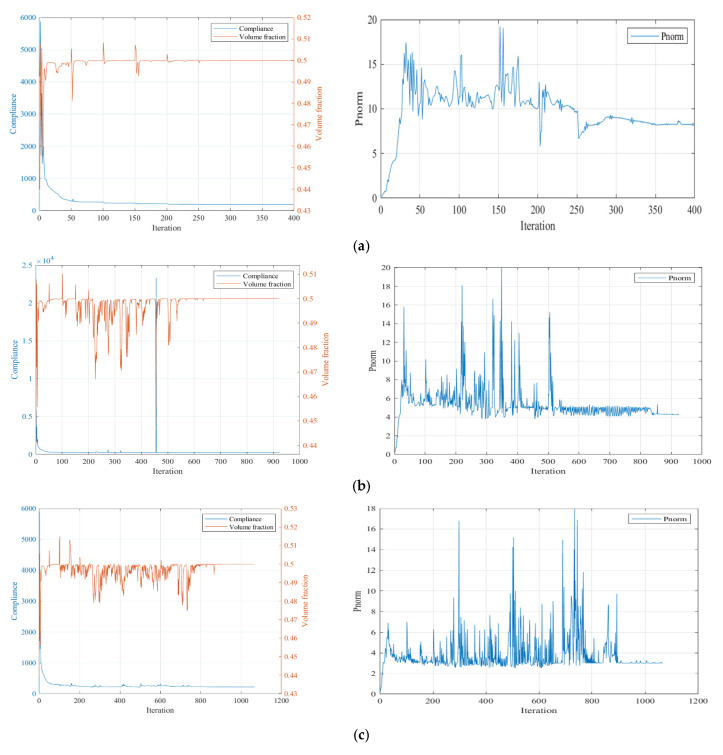
Iteration history curves under different constraint conditions using SIMP interpolation: (**a**) P-norm stress limit set to 10; (**b**) P-norm stress limit set to 5; (**c**) P-norm stress limit set to 3.

**Figure 7 polymers-15-03573-f007:**
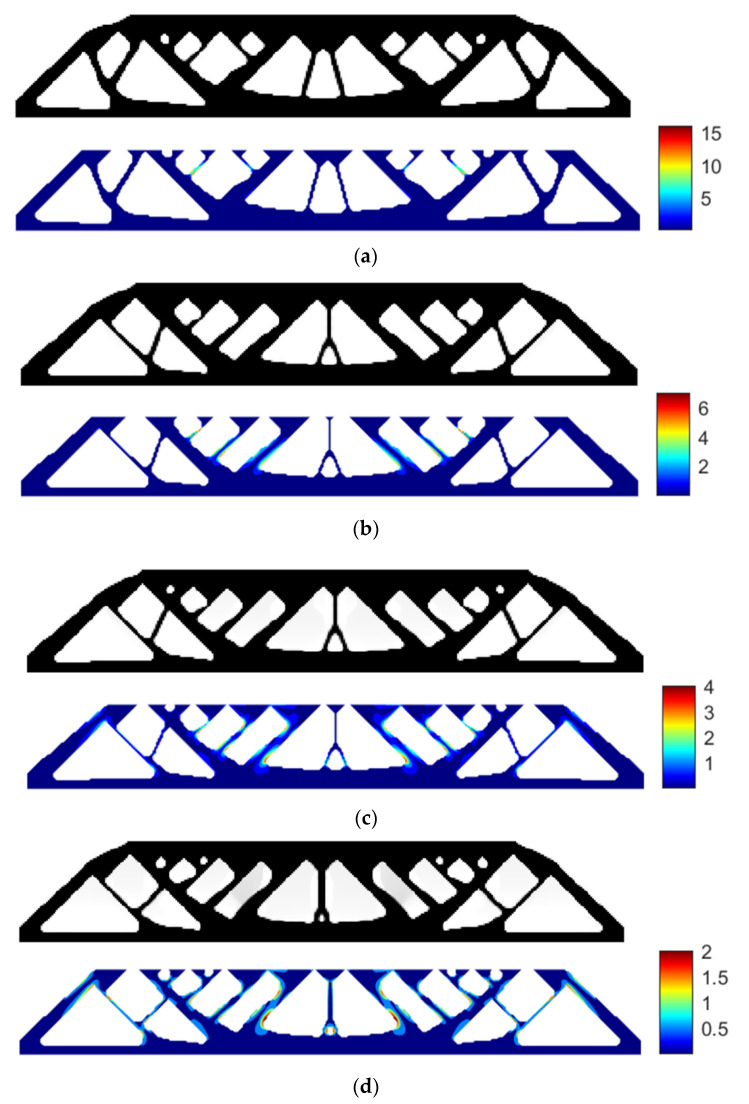
The optimized MBB beam structure and the stress profile plots corresponding to the most stress-concentrated load steps obtained with RAMP interpolation: (**a**) without P-norm stress constraint; (**b**) P-norm stress limit set to 10; (**c**) P-norm stress limit set to 5; (**d**) P-norm stress limit set to 3.

**Figure 8 polymers-15-03573-f008:**
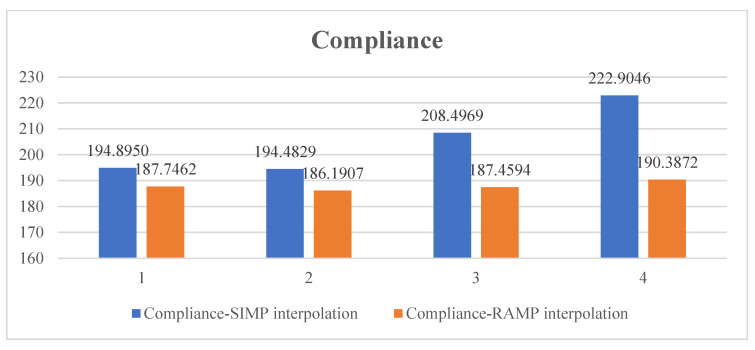
Optimized structural compliances under different constraint conditions and interpolation methods: group 1 represents the topology optimization result without stress constraint; group 2 has the process stress upper limit of 10; group 3 has the process stress upper limit of 5; and group 4 takes the process stress upper limit of 3.

**Figure 9 polymers-15-03573-f009:**
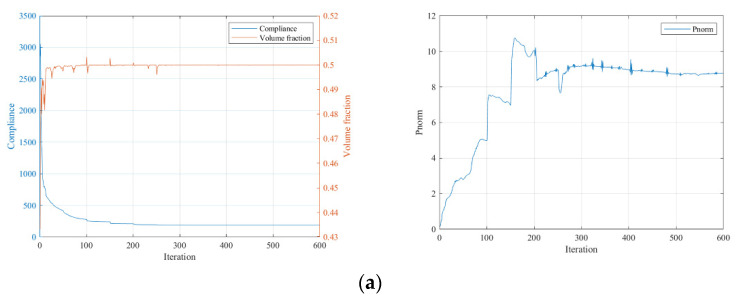
Iteration curves under different constraint conditions using RAMP interpolation: (**a**) P-norm stress limit set to 10; (**b**) P-norm stress limit set to 5; (**c**) P-norm stress limit set to 3.

**Figure 10 polymers-15-03573-f010:**
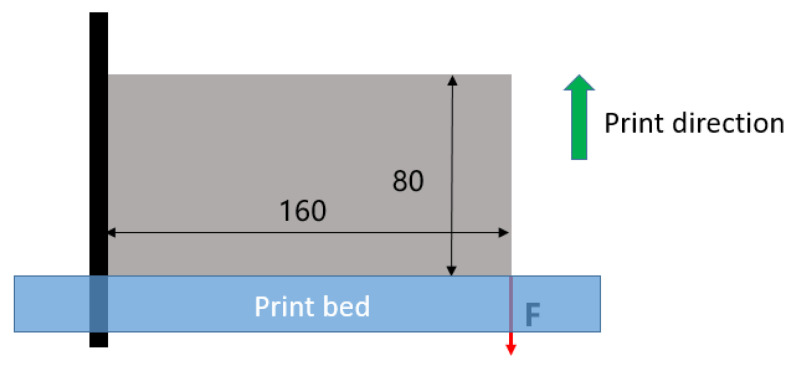
The boundary condition for the cantilever beam.

**Figure 11 polymers-15-03573-f011:**
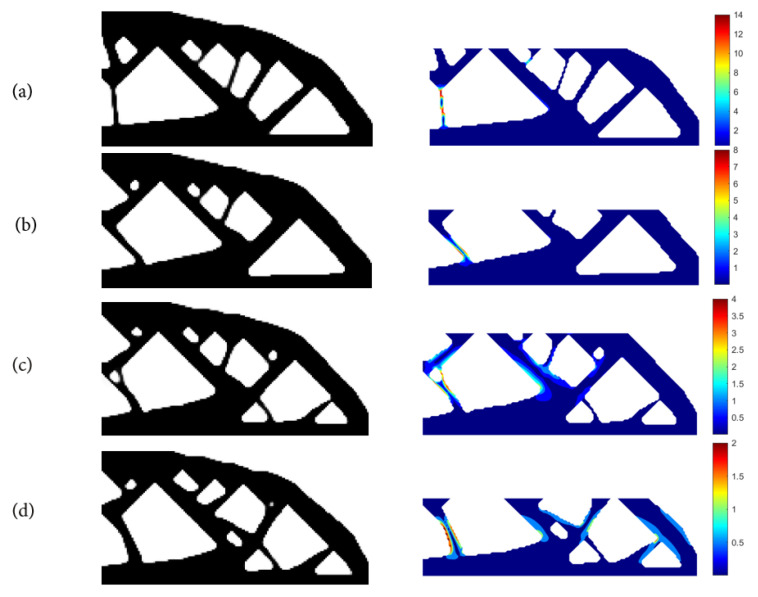
The optimal cantilever beam structure and the stress profiles corresponding to the most stress-concentrated load steps obtained with SIMP interpolation: (**a**) without P-norm stress constraint; (**b**) P-norm stress limit set to 10; (**c**) P-norm stress limit set to 5; (**d**) P-norm stress limit set to 3.

**Figure 12 polymers-15-03573-f012:**
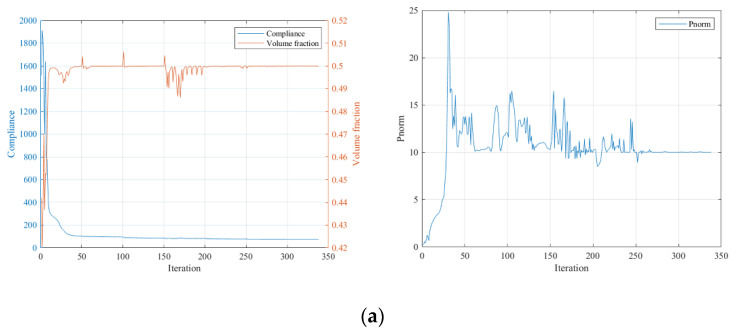
Iteration curves under different constraint conditions using SIMP interpolation: (**a**) P-norm stress limit set to 10; (**b**) P-norm stress limit set to 5; (**c**) P-norm stress limit set to 3.

**Figure 13 polymers-15-03573-f013:**
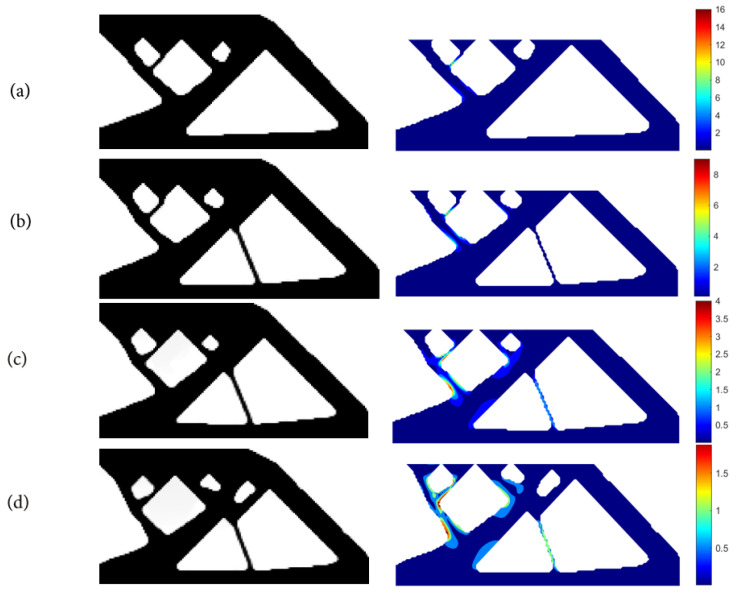
The optimal cantilever beam structure and the stress profiles corresponding to the most stress-concentrated load steps obtained with RAMP interpolation: (**a**) without P-norm stress constraint; (**b**) P-norm stress limit set to 10; (**c**) P-norm stress limit set to 5; (**d**) P-norm stress limit set to 3.

**Figure 14 polymers-15-03573-f014:**
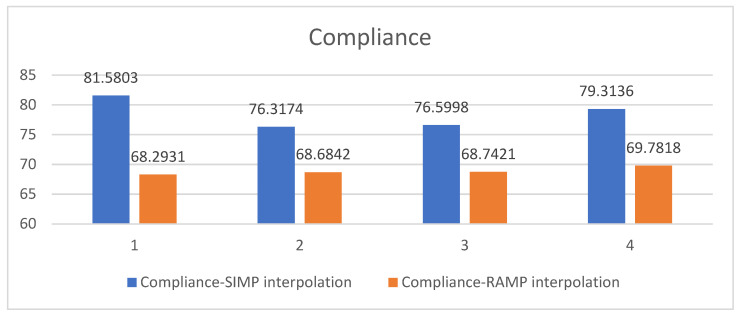
Optimized structural compliances from different constraint conditions and interpolation methods: group 1 represents the topology optimization result without stress constraint; group 2 has the process stress upper limit of 10; group 3 has the process stress upper limit of 5; and group 4 has the process stress upper limit of 3.

**Figure 15 polymers-15-03573-f015:**
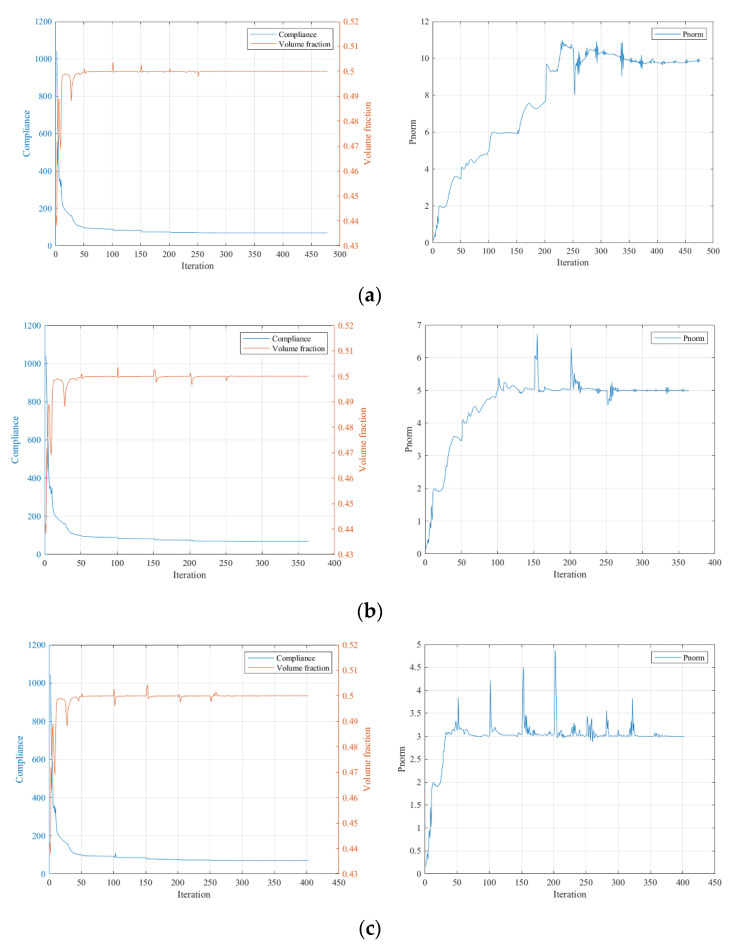
Iteration curves under different stress constraint conditions using RAMP interpolation: (**a**) P-norm stress limit set to 10; (**b**) P-norm stress limit set to 5; (**c**) P-norm stress limit set to 3.

**Figure 16 polymers-15-03573-f016:**
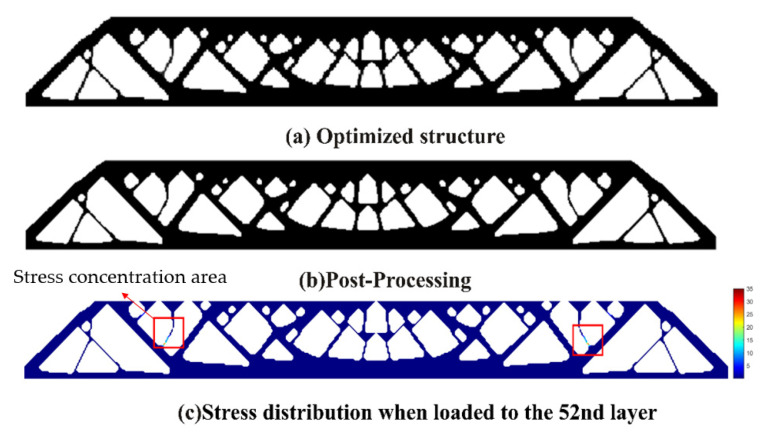
Topology optimization result without process stress constraint: (**a**) optimized MBB beam structure; (**b**) structure after post-processing; (**c**) stress distribution for the most stress-concentrated load step.

**Figure 17 polymers-15-03573-f017:**
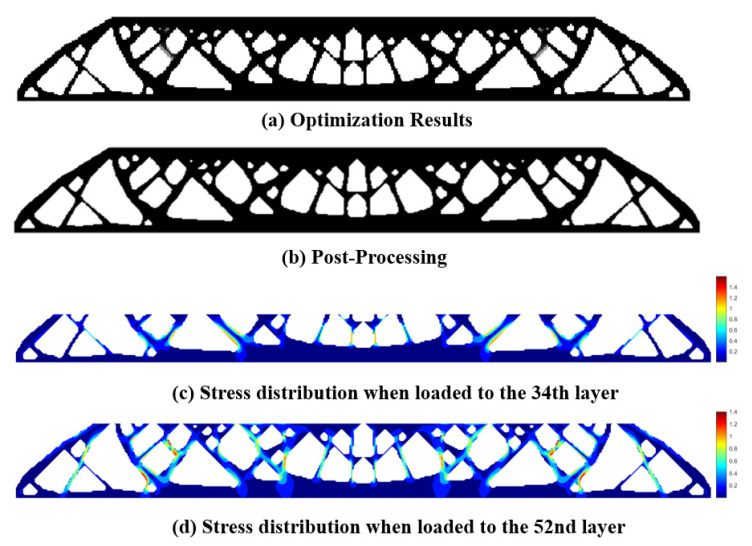
Topology optimization result with process stress constraint: (**a**) optimized MBB beam structure; (**b**) structure after post-processing; (**c**) stress distribution for the most stress-concentrated load step; (**d**) stress distribution for the load case to layer 52.

**Figure 18 polymers-15-03573-f018:**
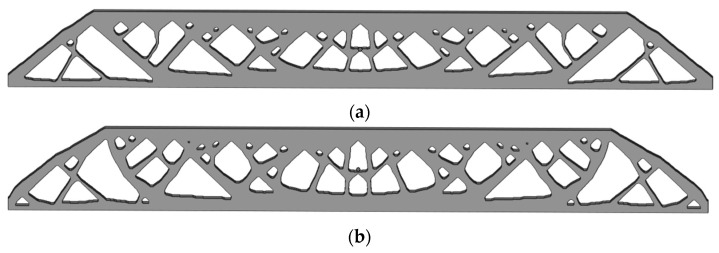
CAD models for 3D printing: (**a**) optimized structure without stress constraint; (**b**) optimized structure with the P-norm stress upper limit of 2.

**Figure 19 polymers-15-03573-f019:**
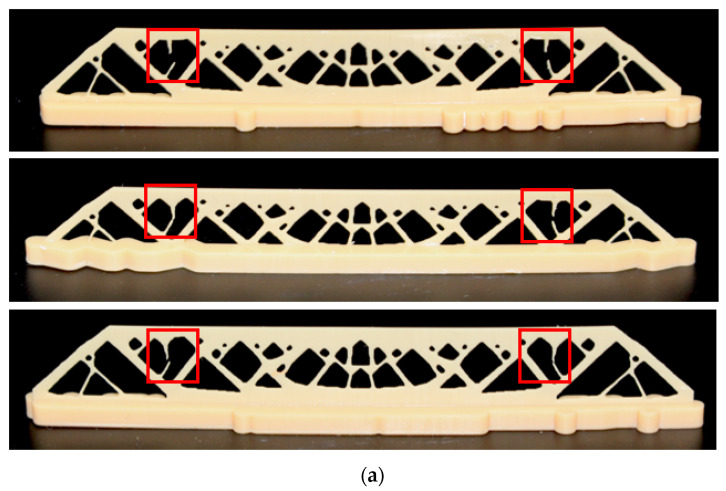
The 3D printing results for (**a**) the optimized design without P-norm stress constraint; and (**b**) the optimized design with the P-norm stress limit of 2.

**Figure 20 polymers-15-03573-f020:**
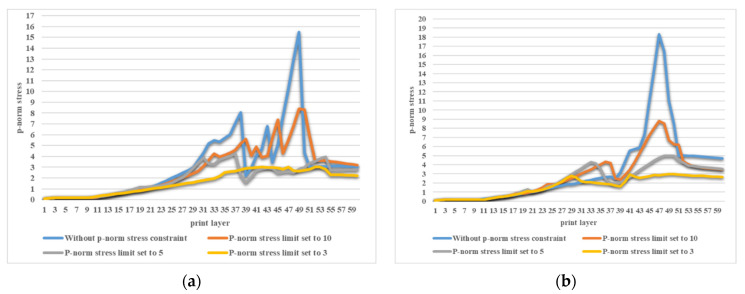
The layerwise P-norm stresses for the MBB beam structure optimized under different constraint conditions: (**a**) using SIMP interpolation; (**b**) using RAMP interpolation.

## Data Availability

Data will be made available on request.

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
