# Peer review of "Topology Optimization for Digital Light Projector Additive Manufacturing Addressing the In-Situ Structural Strength Issue"

_polymers, 2023, doi:10.3390/polym15173573_

Round 1

Reviewer 1 Report

The paper reports a novel process for stress-constrained topology optimization for self-supportimg sructures produced by  DLP additive manufacturing processes. The deduction of the relevant algorithms is clearly presented. The novel approach is validated by the application to several examples of 3D printing and compared to two existing interpolation processes.

This novel process is of high interest for DLP application. The presentation is illstarted by useful diagram^ms and figures

Reviewer 2 Report

This paper proposed a topology optimization method that considers the adhesion-induced stress constraint for DLP printed self-supporting structures. It is an interesting work and is recommended to be accepted for publication after the following revisions.

1. Please explain why the elemental adhesion force is proportional to rho^3 in Eq (2).

2. Please explain how to determine the adhesion coefficient g in Eq (2).

3. Is there only one stress constraint imposed? If the number of stress constraints is the same as that of adhesion layers, please correct the related formulas in Section 3 and 4.1 by adding (i=1, 2, …, m).

4. How does the adhesion layer number m affect the optimized results?

5. From Fig. 1, we can know that vertically downward forces applied to DLP-printed structures include not only adhesion forces but also self-weight gravity loads. Please explain why the proposed method does not consider the gravity load for process-related stress constraints.

6. A few more articles about topology optimization and additive manufacturing should be reviewed here, e.g., Rapid Prototyp. J. 25 (2019) 232–246; Comput. Methods Appl. Mech. Eng. 377 (2021) 113708.

Reviewer 3 Report

The current modeling work is worthwhile to enhance the field of 3D printing but it requires mainly some improved writing, as explained below.

L 31 performs

L 73 Besides topology one also has material design before printing. Please make the reader aware/ On can cite Polymers  2019 11, 1529.

The introduction lacks a general description of complementary modeling efforts, specifically for VAT.

General comment: check size of equations (also at the start green markings are visible in the pdf)

Section 3 needs to be rewritten for a typical reader of Polymers. Either work more with SI or explain better what the approach is vs the state of the art.

Please include in the Results and Discussion more the link with experimental data or at least trends.

General check of grammar.

Round 2

Reviewer 2 Report

This paper has been revised well according to the reviewer's comments. It is recommended to be accepted for publication.

Reviewer 3 Report

The manuscript has sufficiently improved.